# Effects of High-Intensity Interval Training on Inflammatory Biomarkers in Patients with Type 2 Diabetes. A Systematic Review

**DOI:** 10.3390/ijerph182312644

**Published:** 2021-11-30

**Authors:** José Manuel Leiva-Valderrama, Adrián Montes-de-Oca-Garcia, Edgardo Opazo-Diaz, Jesus G. Ponce-Gonzalez, Guadalupe Molina-Torres, Daniel Velázquez-Díaz, Alejandro Galán-Mercant

**Affiliations:** 1Faculty of Nursing and Physiotherapy, Puerta del Mar University Hospital, University of Cadiz, 11009 Cadiz, Spain; jose.leivavalderrama@alum.uca.es (J.M.L.-V.); alejandro.galan@uca.es (A.G.-M.); 2MOVE-IT Research Group, Department of Physical Education, Faculty of Education Sciences, University of Cadiz, 11519 Cadiz, Spain; adrian.montesdeoca@uca.es (A.M.-d.-O.-G.); edopazo@uchile.cl (E.O.-D.); jesusgustavo.ponce@uca.es (J.G.P.-G.); 3Biomedical Research and Innovation Institute of Cadiz (INiBICA) Research Unit, Puerta del Mar University Hospital, University of Cadiz, 11009 Cadiz, Spain; 4Department of Physical Therapy, Faculty of Medicine, University of Chile, 8380453 Santiago, Chile; 5Department of Nursing, Physiotherapy and Medicine, Faculty of Health Sciences, University of Almería, 04120 Almería, Spain; guada.lupe@ual.es

**Keywords:** HIIT, T2D, inflammatory biomarkers

## Abstract

Background: Due to the prevalence and incidence worldwide of type 2 diabetes, and the significant role physical activity plays in these patients, a systematic review has been conducted to find out the effects that high-intensity interval training has on inflammatory biomarkers in subjects with type 2 diabetes. This project aims to determine the effect this training modality has on inflammatory biomarkers, in addition to observing its effects on the values of body composition and determining if this is a more effective, less effective or equally effective alternative to standard aerobic or resistance training. Methods: A search was conducted in the months of November and December 2020 on different databases: Pubmed, WoS and PEDro. A protocol for this systematic review was registered in PROSPERO (Registration number: CRD42021281186). The studies selected met the previously defined inclusion criteria, and the methodological quality of the papers used was evaluated according to the Downs and Black Checklist. Results: Out of 46 studies found, seven were included. The most relevant data concerning the characteristics of the clinical trials and HIIT characteristics, the values of body composition and the biomarkers under study were extracted from each study. Moreover, the results obtained from the different studies were described. Conclusions: HIIT could have an effect on inflammatory biomarkers. There is likely to be a relationship between changes in inflammatory profile and fat loss. A controlled diet may be a good complement to reduce the inflammatory profile. Further studies are required to determine whether HIIT is a better, worse or an equivalent alternative to medium-intensity aerobic exercise to improve the inflammatory profile.

## 1. Introduction

Type 2 diabetes (T2D) is a metabolic disease characterised by a failure in the signalling cascade of the insulin hormone, preventing the partial or total capture of glucose inside body cells, which leads to what is called insulin resistance [1]. Therefore, although the body secretes insulin, this disease implies this cannot properly perform its function, keeping glycaemia levels high. At advanced stages of the disease, a disorder in the pancreatic β cells may even occur, which is characterised by an insulin deficiency. According to the International Diabetes Federation, the number of people with diabetes will reach 629 million in 24 years (T2D accounting for more than 90% of diabetes cases) due to its increasing prevalence and incidence worldwide [2,3].

T2D mainly occurs due to two reasons—increased adiposity/obesity and decreased physical activity levels—two factors showing a more and more significant prevalence among the population [4]. Multiple studies have demonstrated the relationship between physical inactivity, obesity and the increased prevalence of type 2 diabetes [5,6,7,8]. For this reason studies advocate the need for physical exercise and a healthy diet in the prevention of T2D [2,9].

Nowadays, a regular physical activity implemented in our daily life is proved to increase our quality of life, both in healthy people and in people suffering from any kind of health disorder or pathology. Systematic reviews have shown significant findings between physical exercise and an improvement in glycaemic control and insulin response [10], in addition to the anti-inflammatory and fat reduction effects provided by regular physical activity [11]. However, the lack of time is still the main reason leading to a lack of adherence to exercise, so new physical exercise tools have been developed to reduce the session duration as it is the case of High-Intensity Interval Training (HIIT) [12].

This training modality consists of alternating repeated short or long periods of high intensity exercise with recovery periods, reaching at least 90% of maximum oxygen consumption (VO2max) in working periods. Nowadays, this is one of the most popular and effective training modalities for improving both at cardiorespiratory and metabolic levels [13]; however, compared to traditional continuous aerobic training, which is characterized by exercise sessions of long duration at moderate intensities, HIIT appears also to be an effective tool to improve health but in shorter sessions [12,14]. Nevertheless, there is no clear evidence on the benefits of this type of training for inflammatory markers in T2D as a health sign in such pathology, which exposes the need for this systematic review.

Indeed, multiple studies have demonstrated the relationship existing between T2D, obesity and an inflammatory profile due to the presence of some biomarkers associated with inflammation [15,16,17,18,19,20].

This inflammation is a protective response against pathogens, toxins, etc. This inflammatory process may occasionally end up damaging the human body. Over the last few years, some studies have supported obesity as an inflammatory disorder in the body, in addition to being associated with other types of disorder such as the metabolic syndrome and type 2 diabetes (insulin resistance effect) [20].

A previous study [20] showed the relationship between inflammatory biomarkers and obesity, insulin resistance and cardiovascular disease conditions and includes the scientific papers in which these findings are outlined. There we are shown how the IL-6 and the IL-18 keep their pro-inflammatory functions, on the other hand, the TNF- α takes the role of mediator in the inflammatory/immune response and high levels of CRP are found as an inflammatory response [20]. Moreover, the study shows the appearance of high levels of inflammatory biomarkers in mice with diabetes, obesity or chronic inflammation, with insulin resistance effects (IL-6, TNF-α) appearing in greater levels in obese individuals and pro-inflammatory effects (IL-1, CRP), which also appear in greater levels in obese subjects [21]. Likewise, in mice, some inflammatory biomarkers such as the TNF- α is more relevant in the adipose tissue of persons suffering from obesity [16], and in case of being high at the systemic level, it is known as a major inhibitor of insulin signalling, mainly in the liver and muscles causing an insulin resistance [22]. Other studies also show the existing relationship between high concentrations of IL-6 and CRP and the risk of suffering from T2D [15,17,18,23]. In relation to the IL-8, it has been proved how patients with T2D show high circulating values of this biomarker associated with a worse inflammatory condition and metabolic control [24]. With regard to the IL-10, low levels of this biomarker are present in patients with T2D compared with healthy patients, which leads to a lesser control or prevention in the production of biomarkers such as the TNF- α and IL-6 [25]. Other studies show an increase in sCD163, which is closely linked with the activation of macrophages and the risk of suffering from T2D [26,27]. Concerning the IL-1Ra, low levels appear in patients with T2D [28,29]. Furthermore, a 2009 study demonstrated that the treatment with IL-1Ra in patients with T2D reduced inflammatory biomarkers and improved the functioning of β cells [30]. As for the IL-15, there are few studies addressing this inflammatory biomarker and T2D, but there are some concerning the presence of high levels of this biomarker in patients with T1D [31,32].

It is interesting to consider also other biomarkers, such as irisin and some adipokines like adiponectin, leptin and resistin. Some studies have proved the close relationship existing between irisin, T2D and obesity. There are low concentrations of this biomarker in patients with T2D, showing a positive relationship between this biomarker and BMI, insulin sensitivity and weight loss, so this could play a key role in glucose intolerance existing in T2D [33]. Concerning the adipokines, the function of both leptin and resistin is pro-inflammatory, they take part in the immune response, activate macrophages and lymphocytes [27]. It is increased in obese people favouring autoimmune processes, in addition to body inflammation [27]. Adiponectin function is anti-inflammatory; however, other biomarkers including TNF and leptin reduce its levels [27]. It has an anti-inflammatory function favouring the production of IL-1Ra and IL-10. Both in obese and diabetic persons it is found in low levels, which leads to an inflammation at the systemic level and other issues [27].

There are other inflammatory biomarkers less frequently addressed in the studies used for this systematic review, such as FGF21, SPARC and ANGPTL4. The FGF21 seems to improve hyperglycaemia and obesity conditions when administered to diabetic and obese patients for therapeutic purposes [34]. Regarding the biomarker SPARC, some papers show a potential relationship between an increase of this biomarker and the pathologic mechanism of obese and type 2 diabetic patients [35]. The ANGPTL4 is found at high levels in patients with diabetes, and it is connected with dysfunctions of the adipose tissue and inflammation [36].

When examining the subjects’ inflammatory profile, some of the most studied and most relevant inflammatory biomarkers are mainly produced by adipocytes, such as the TNF- α and the IL-6 [20]. Based on this, we could think the inflammatory profile of people with T2D would be closely linked with the subjects’ fat profile.

Regular exercise exerts an anti-inflammatory chronic effect, diminishing the plasmatic markers of inflammation [37]. The elevation of circulating anti-inflammatory biomarkers after an acute bout of exercise seems to be dependent on intensity, with a greater response after HIIT compared to moderate-intensity continuous exercise [38,39]. The HIIT relies more heavily on carbohydrate oxidation than on endurance training, which has been related with a blunted acute inflammatory response [40,41]. However, this finding has not consistently been shown in the literature [42,43].

HIIT as a modality of exercise and an alternative to the traditional aerobic and resistance training has been booming lately [44,45,46]. However, the truth is that little information is found in databases on the effects this modality of exercise has on inflammatory biomarkers. Based on this, we have decided to focus this systematic review on looking at the therapeutic effect a modality of physical exercise may have on patients with T2D. Therefore, we have compiled the clinical trials studying T2D patients and the effects this modality of exercise could have on inflammatory biomarkers to draw a conclusion and determine the existing relationship between HIIT, T2D and inflammatory biomarkers.

### 1.1. Approach to the Problem

Given that T2D is a very prevailing pathology and that physical exercise has been proven to have positive effects on the profile of diabetic patients by reducing insulin resistance and improving the inflammatory role of this disease, by doing this systematic review we seek to verify what the effects of HIIT as a modality of physical exercise has on inflammatory biomarkers in patients with T2D and if it is an equally, less or more effective alternative to aerobic or resistance exercise training.

### 1.2. Objectives of the Study

The main objective of this systematic review consists of analysing the effect HIIT has on inflammatory biomarkers in patients with T2D. The secondary objectives would be to find an answer to the following questions: which modality of HIIT is more effective to improve the inflammatory profile; what effect HIIT has on the values of body composition and, finally, what are the different effects HIIT and aerobic or resistance exercise have on inflammatory biomarkers and the values of body composition.

## 2. Materials and Methods

This systematic review has been conducted following the PRISMA [47] model and its various items, a guide created to serve as a protocol and improve the quality of systematic reviews and meta-analyses. Moreover, a protocol for this systematic review was registered in PROSPERO (Registration number: CRD42021281186).

### 2.1. Search Strategy

In order to prepare this systematic review, the PICO question was the search strategy selected for which the kind of patient, the intervention to be studied and the variables to be studied were considered for approaching the problem.

P: Diabetes Mellitus Type 2.I: High-Intensity Interval Training.C: Alternative treatment or control.O: Values of body composition and inflammatory biomarkers.

A systematic search of clinical trials was launched on the Internet on 25th November 2020 in the Pubmed, WOS and PEDro platforms and was completed by 2nd December 2020. The search engines used in the different databases were the following:

Pubmed: ((“high-intensity interval training” [Title/Abstract] OR high-intensity interval training [MeSH] OR “HIIT” [Title/Abstract] OR “High Intensity Interval Training” [Title/Abstract]) AND (Diabetes Mellitus, Type 2 [Mesh] OR “Diabetes Mellitus Type 2” [Title/Abstract]) AND (Cytokines [MeSH] OR cytokines [Title/Abstract] OR Interleukins [MeSH] OR Interleukins [Title/Abstract] OR Monokines [MeSH] OR Monokines [Title/Abstract] OR Tumor Necrosis Factors [MeSH] OR “Tumor Necrosis Factors” [Title/Abstract]))

WOS: TS=((“High-Intensity Interval Training*” OR “Interval Training, High-Intensity*” OR “Training, High-Intensity Interval*” OR “Exercise, High-Intensity Intermittent*” OR “High-Intensity Intermittent Exercise*” OR “Sprint Interval Training*” OR “high-intensity interval”) AND (“Diabetes Mellitus, Noninsulin-Dependent” OR “Diabetes Mellitus, Ketosis-Resistant” OR “Diabetes Mellitus, Ketosis Resistant” OR “Ketosis-Resistant Diabetes Mellitus” OR “Diabetes Mellitus, NonInsulin Dependent” OR “Diabetes Mellitus, Non-Insulin-Dependent” OR “Non-Insulin-Dependent Diabetes Mellitus” OR “Diabetes Mellitus, Stable” OR “Stable Diabetes Mellitus” OR “Diabetes Mellitus, Type II” OR “NIDD” OR “Diabetes Mellitus, Noninsulin Dependent” OR “Diabetes Mellitus, Maturity-Onset” OR “Diabetes Mellitus, Maturity Onset” OR “Maturity-Onset Diabetes Mellitus” OR “Maturity Onset Diabetes Mellitus” OR “MODY” OR “Diabetes Mellitus, SlowOnset” OR “Diabetes Mellitus, Slow Onset” OR “Slow-Onset Diabetes Mellitus” OR “Type 2 Diabetes Mellitus” OR “Noninsulin-Dependent Diabetes Mellitus” OR “Noninsulin Dependent Diabetes Mellitus” OR “Maturity-Onset Diabetes” OR “Diabetes, Maturity-Onset” OR “Maturity Onset Diabetes” OR “Type 2 Diabetes” OR “Diabetes, Type 2” OR “Diabetes Mellitus, AdultOnset” OR “Adult-Onset Diabetes Mellitus” OR “Diabetes Mellitus, Adult Onset”) AND (“Cytokines” OR “Interleukins” OR “Monokines”OR “Tumor Necrosis Factors” OR “Necrosis Factors, Tumor” OR “TNF Receptor Ligands” OR “Receptor Ligands, TNF” OR “Tumor Necrosis Factor Superfamily Ligands” OR “TNF”)) Timespan: All years. Indexes: SCI-EXPANDED, SSCI, A&HCI, CPCI-S, CPCI-SSH, BKCI-S, BKCI-SSH, ESCI, CCR-EXPANDED, IC.

PEDro: Diabetes type 2 and HIIT

### 2.2. Selection Criteria

Papers were considered eligible for this systematic review if they met the following inclusion criteria:

The paper had to address a randomised or non-randomised clinical trial.

Subjects have to suffer from Type II Diabetes Mellitus.

Papers had to include an intervention group in which HIIT is used as a modality of training/intervention.

Papers had to include measurements of body composition such as size, BMI, weight, perimeters, relationship indices, as well as other features related to bioimpedance analysis.

Papers had to include at least one inflammatory biomarker such as, for example, IL-6, TNF-α, CRP, or IL8.

Studies had to be conducted involving human subjects.

### 2.3. Paper Selection and Data Abstraction

Once the search was completed, the results of it were evaluated independently. For this purpose, Rayyan, a digital resource was used [48]. To begin with, a first screening was made based on title and abstract by author and evaluator (JMLV and AGM). Subsequently, a screening was made by carefully reading the rest of the papers to check if they met the inclusion criteria. All the screenings were blind one from another for later discussion in case no agreement was reached. If no agreement was reached, a second evaluator was requested to include the paper or not.

### 2.4. Quality Assessment

Papers meeting the inclusion criteria were examined to find out their quality. For this purpose the “Downs and Black Checklist” [49] was used. This was modified by removing the last item “power” scoring the statistical power, so that the maximum score to be assigned to each article was 26; thus, the evaluation encompassed the reporting strength, and the internal and external validity. This scale was used to evaluate the quality of the papers because it was created to evaluate the quality of randomised and non-randomised clinical trials.

## 3. Results

### 3.1. Selection of Studies

A total of 46 clinical trials were obtained, out of which 39 were excluded (nine for being duplicated, 26, by title and abstract, and four for not meeting the inclusion criteria), leaving a total of seven papers included. In order to facilitate the understanding and follow-up of this systematic review, a flow chart was created showing the different stages completed for the selection of the studies (Figure 1).

### 3.2. Evaluation of the Quality of the Studies

Out of the seven clinical trials used, the one ranking the highest as per the methodological evaluation scale reached 21 points out of 26 [50]. The one with the lowest score got 11 points out of 26 [51]. The remaining papers got a score between 16 and 18 out of 26 [28,52,53,54,55]. Full assessment of the studies used may be checked in the relevant Table 1.

### 3.3. Data Abstraction

Based on the need to thoroughly compare the seven clinical trials used in this systematic review [28,50,51,52,53,54,55], for comparison purposes a summary table was prepared stating the different relevant data provided by each study. In order to facilitate the reading and understanding of the results, the information was divided into 3 tables of results. Firstly, a table showing the general characteristics (Table 2), in which the seven studies and the relevant data to be studied may be checked—year, country, methodological quality score, kind of study, kind of patients, length of the intervention, women and men ratio, age range, weight conditions, level of physical activity, groups to be studied and intervention in every group. Secondly, another table indicates the relevant data for the HIIT intervention, measurements of body composition and biomarkers analysed in the respective studies (Table 3). The last table shows the results obtained from each of the studies, by indicating the presence or non-presence of significant changes in the variables under study (Table 4). Additional variables and complementary information can be found in the Appendix A (Appendix A).

Once the papers had been analysed and the most relevant data extracted, for every trial the type of treatment with the modality of therapeutic exercise was examined. Since the term HIIT encompasses multiple variations when implemented, a “Classification of HIIT protocols” [44] was used to consider the different variations of this training modality. This classification includes the following variations: long-interval [LI-HIIT], moderate-interval [MI-HIIT], short-interval [SI-HIIT], sprint-interval [SIT] and repeated-sprint [RST]), session volumes (high-volume [HV-HIIT], moderate-volume [MV-HIIT] and low-volume [LV-HIIT]) and training periods (long-term [LT-HIIT], moderate-term [MT-HIIT] and short-term [ST-HIIT]) (Table 5).

### 3.4. General Description of Data Obtained

Studies involved patients with T2D except for two of them, which included both diabetic patients and healthy subjects [51,54]. The total patients considered in this review were 161 (HIIT = 161, MCT = 38 and Control = 59). The age range of the participants in these clinical trials of the papers used was between 30 and 75 years of age. Only four out of the seven studies stated the ratio of men and women participating in such studies [28,50,52,54]. Only one of the studies did not provide the patients’ BMI data at the beginning of the intervention in the respective studies [54]. Considering the six studies that indicated these data, the BMI of all subjects was over ≥25kg/m² and BMI < 48kg/m². With respect to the length of the treatment in these studies, only periods in which training was monitored were considered, since some studies include periods in which training is conducted at the patients’ home without supervision. All the HIIT interventions were conducted on a cycle ergometer and treadmill.

### 3.5. Characteristics of HIIT, Values of Body Composition and Biomarkers

In relation to the characteristics of HIIT, an intensity level of 90% HRmax was reached in every trial, except for one of them, which does not specify the intensity for the maximum effort [54]. One of the studies shows HIIT intensity as all out [28]. All clinical trials used three days per week for the intervention, the study with the major duration lasting for 24 weeks and the shortest being of 4 weeks. The shortest HIIT working out time was 30 s [28], while the longest reached 240 s [52]. Most recurrent HIIT modalities were LI-HIIT, HV-HIIT and LT-HIIT.

In these studies, the biomarker most frequently examined was TNF-α in 6 of 7 studies, 3 studies showed a significant decrease and 3 no change in this parameter. The second most examined parameter was IL-6 in 5 of 7 studies and showed a significant decrease in 3 studies. Moreover, the most frequently used variable of body composition was the BMI.

## 4. Discussion

This systematic review was conducted for finding out an answer to the effect HIIT as a modality of therapeutic exercise has on inflammatory biomarkers in patients with T2D. As described above, this disease is normally associated to obesity and the absence of exercise implemented in the lifestyle of the diabetic subject.

The heterogeneity of the papers used for this systematic review is due to the little number of studies existing nowadays and addressing the question made for the purposes of this work. As a result of the outcomes in these studies and their diversity, the results are not very conclusive when comparing one to another. The discussion on the main findings of the studies included in this systematic review is found below.

### 4.1. HIIT and Inflammatory Biomarkers

First, we should highlight that the sizes of the various samples were quite small. If we observe the participants who only did HIIT, the number of subjects substantially diminishes 13/51 [50], 16/36 [52], 14/42 [28], 39/39 [55], 8/14 [53], 10/23 [54] and 49/80 [51] and if we focus on those doing HIIT and suffering from T2D the number is even smaller.

Secondly, we are thoroughly analysing the different components of HIIT and factors impacting on it, and comparing which outcomes were recorded for each of them.

If we consider HIIT intensity, studies did not include an intensity of HRmax lower than 70% or higher than 95%. All the interventions were conducted within this small range, the highest intensities (90%–95%) being attained gradually as the study was conducted. Thus, we could say that because of the heterogeneous outcomes obtained in the different studies, in this case, the exercise intensity was not directly responsible for the results obtained concerning biomarkers and body composition, as we have some studies showing significant results and others with no significant results, and all of them ranged within this small intensity interval, attaining at the end 90%–96% of HRmax or all out.

If we focus on the number of series made during the training days, they are very different when comparing all the studies. Studies not obtaining significant outcomes for inflammatory biomarkers in patients with T2D used a number series of one [50], four [52], or ten [54]. Studies that did obtain significant outcomes used a number of series of four [28], ten [55], five [53] and six to 12 [51]. As described, it is complicated to determine the number of series which may have had a significant impact on biomarkers, as studies using the same number of series show different results—this is probably due to the starting level and the heterogeneous samples. We cannot determine either if there is a direct relationship between the number of series and the significant results. According to the data we are handling in this systematic review, the number of series is not likely to be decisive for the results if examined separately. For this reason, we need to examine this bearing in mind the days and weeks spent in each study for treatment implementation. Taking a closer look to it, we have the advantage that every study implemented exercise for three days a week, the difference lying therefore in the number of weeks. The study with the longest duration for treatment implementation took 52 weeks and it was the one by Magalhães et al. [50] and it did not obtain significant results for any biomarker within the HIIT group. Another study was conducted for 24 weeks [55] and did obtain significant results for several biomarkers. The study with the shortest duration took four weeks [53] and it did find significant changes in one of the biomarkers. The rest of the studies extended from 8 to 12 weeks [28,51,52,54] and 2 of them did not obtain any significant result for the HIIT groups with T2D [52,54]. Based on this, and according to the current studies, we could say that we have not found any direct relationship between the duration and the presence of significant results. It is likely to be more connected with the HIIT working out time and the number of series completed in each session, since the study with the longest duration only conducted one series per session and it did not obtain any results [50]. The rest of the studies, except for two of them [52,54], used intervals between 4 and 12 series and working out times between 30 s and 240 s and they did find significant changes in patients with T2D doing HIIT.

However, considering the data we handled and the very few studies available nowadays, it is difficult to determine which type and characteristics of HIIT are able to drive significant changes in inflammatory biomarkers in patients with T2D.

In this systematic review, it should be pointed out that one of the studies which really found significant differences after treatment and also between different groups was completely made up of women with T2D [28]. This might lead us to think that future papers should come up addressing the effects HIIT has on inflammatory biomarkers and the values of body composition for different populations of men and women. The truth is that one of the studies detailed values for women and men separately, but in this case no significant results were obtained in any of the groups [52]. Maybe because the HIIT characteristics did not trigger the expected effects on patients.

### 4.2. HIIT and Values of Body Composition

This search for significant changes could not only depend on HIIT characteristics and variations, but also the characteristics of the patients under study and their type of profile are likely to have something to say. The effects linking HIIT and the values of body composition are discussed below.

In this systematic review, only three studies out of a total of seven provide pre- and post-treatment data for the different variables of body composition analysed in each study [28,53,54]. For comparison purposes, it would have been interesting to have all the pre- and post-treatment values for all seven studies. HIIT groups with T2D showed significant changes in BMI, weight, LBM and abdominal fat mass [28,54]. The paper by Banitalebi E et al. [28] shows only that there were significant differences between groups; the study by Madsen SM et al. [54] did show a significant reduction in the values already mentioned for the HIIT group with T2D, but the study by Dünnwald T et al. [53] interestingly showed a significant increase in the fat free mass value within the CMT group; moreover, the HIIT group with T2D did not record changes in the body composition variables measured in such a study, not like in the CMT group. Except for the study by Madsen SM et al. [54], which did not provide the pre-treatment BMI values, patients had overweight BMI ≥ 25kg/m^2^ or obesity BMI ≥ 30kg/m2 based on the standard values for overweight and obesity laid down by the World Health Organisation (WHO) [56]. These data on BMI lead to us thinking about the relationship between body composition and biomarkers.

### 4.3. Values of Body Composition and Biomarkers

It would be interesting to gather information on the pre- and post-treatment values of body composition to determine if there is a certain relationship between recording significant changes in inflammatory biomarkers and the presence of significant changes in the values of body composition. If we had all these data, we could maybe observe that for having significant changes in the inflammatory biomarkers, the physical exercise implemented—in this case HIIT—needs to trigger first a significant reduction in body fat. For example, in the study by Afrasyabi et al. [51], the group recording the most relevant reduction in TNF-α was the group of obese, diabetic subjects under an HIIT intervention. This relationship is observed in studies such as the paper by Pedersen et al. [57] in which a significant reduction in TNF-α and CRP did take place in patients losing weight after a year of aerobic exercise. Furthermore, in other studies, like that in the paper by G. Zoppini et al. [58], they showed that no change was recorded in the biomarkers TNF-α and CRP in patients who did not lose weight during a moderate intensity aerobic training program. However, in contrast to this, one of the studies used in this systematic review showed a significant reduction in the values of weight and abdominal fat within the HIIT group and T2D, but it did not show changes in any of the inflammatory biomarkers [54], and another study showed a significant increase in irisin within the HIIT group, but not in the values of body composition [53]. The results of the latter study are interesting, as irisin is in charge of converting white fat to brown fat [33].

Among other issues, due to the very few studies available, it is quite easy for body composition variables not matching with the inflammatory biomarkers studied, and this makes it difficult to determine the existing relationships since, if other biomarkers had been examined, maybe significant changes would have come out.

### 4.4. HIIT, Biomarkers and Diet

As we have introduced in this systematic review, lifestyle is critical for the prevention and treatment of patients with T2D. Lifestyle consists of both a regular physical activity and a healthy diet, considering a healthy diet is a diet according with recommendations from authoritative organizations [59]. Although more long-term intervention trials are needed, Papamichou et al. [60] have shown the effect that dietary patterns of a healthy diet should be implemented in public health strategies in order to better control glycaemic markers in patients with T2D. This is why having information on and controlling the diets the subjects were on would help to observe the effects it may have. The low-carbohydrate high-fat diet is a re-emerged dietary approach, which is characterized by a decreased carbohydrate intake and high levels of fat consumption with adequate protein provided [61]. A recent systematic review and meta-analysis has shown that a low-carbohydrate high-fat diet combined with HIIT reduces body weight and fat mass, while maintaining lean body mass and enhancing aerobic capacity. Moreover, another recent systematic review and meta-analysis has shown that patients who adhere to a low-carbohydrate diet for six months can experience remission of diabetes without adverse consequences [62]; however, these studies did not included inflammatory biomarker analyses [63]. In the papers selected for this systematic review, only one of the studies showed the effects a HIIT training that four different types of diets have on patients with T2D [55]. In this clinical trial, it was determined that low-fat and low-carbohydrate diets, along with HIIT, had major effects on the reduction of the levels of biomarkers IL-6, TNF-α, leptin and resistin. Based on a single study, we cannot advocate that a type of diet along with HIIT training is the key for the reduction of inflammatory biomarkers. However, it would be interesting to investigate which diets the subjects of available studies were on, so that they could be compared, as there is likely to be a close relationship with the effect they may have on these biomarkers. Furthermore, in relation to the diet implemented in the different groups, the BMI of the participants should be also highlighted as it ranged between 30 and 39 kg/m^2^, these BMI falling into the category of obesity according to the WHO [56]. Thus, we may repeat that the potential effect on the inflammatory biomarkers might be closely linked with fat loss. If the pre- and post-treatment data had been provided for these persons, after the HIIT treatment and a sort of controlled diet, they are likely to have experienced changes in their fat composition.

### 4.5. Effects on Inflammatory Biomarkers of the Different Intervention Modalities

If we take a closer look to the studies that obtained significant changes within the T2D groups under a HIIT intervention, a reduction in the values of IL-6, IL-15, TNF-α, resistin and leptin was obtained [28,55]. Conversely, an increase in the values of FGF21, irisin and adiponectin was recorded [53,55]. Based on the functions of these biomarkers which are outlined in the introduction herein, changes recorded in these studies are connected to an anti-inflammatory response of the body, as the pro-inflammatory agents are reduced, and the anti-inflammatory agents are increased. We have to point out that in the study by Banitalebi E et al. [28], the calculations for statistical purposes are made using ANCOVA to determine which training group leads to more significant changes in biomarkers and which significant differences there are among the various groups, but afterwards they do not specify these details when answering these questions, they provide the main differences between pre- and post- based on which we can infer the results, but not certainly determining which group is more significant. Therefore, we also have to consider that within groups with an alternative treatment a significant reduction was obtained in biomarkers such as IL-6, IL-15, [28] and oddly enough in the study by Magalhães et al. [50] only the IL-6 obtained a significant reduction within the MCT group compared with the control group. With respect to the study by Madsen SM et al. [54], only the IL-1Ra proved an increase within the HIIT group of nondiabetic patients doing HIIT.

The HIIT triggered an anti-inflammatory effect on patients with T2D in four out of the seven clinical trials used [28,51,53,55]. Out of these four, two of them [28,53] are compared with other groups of patients with T2D under an alternative physical treatment. The other two [51,55] do not include this comparison. In one of these [55] groups, they only differ one from another by the type of diet and the other [51] is compared with HIIT groups of nondiabetic patients or control groups, but it does not provide an alternative physical treatment.

Out of the seven clinical trials used in this review, four of them [28,50,52,53] show an alternative treatment for patients with T2D, which is the moderate intensity aerobic training. In two of them [28,50], this type of treatment resulted in the anti-inflammatory effect.

In order to determine which type of physical treatment obtains more frequently an anti-inflammatory effect, it would be fair to use only the four studies of this systematic review [28,50,52,53], in which a HIIT group and an aerobic exercise group were included. Bearing this in mind, we may not determine which the best alternative is to trigger an anti-inflammatory effect in patients with T2D, since out of these four studies, HIIT only got significant outcomes in two of them [28,53], likewise the moderate intensity aerobic training which showed significant results in two of them [28,50].

### 4.6. Limitations of the Study

When conducting this review we found several limitations. First, the limited number of clinical trials found using HIIT as a treatment method, and analysing the effects on some inflammatory biomarkers.

For this same reason, another current limitation is the absence of homogeneity concerning the types of HIIT and biomarkers addressed in each of them, which makes comparison more difficult.

Another limitation is that most of the studies included that met the inclusion criteria showed only pre- and post-results of biomarkers but not of the values of body composition.

Three of the studies used for this systematic review differed from the other four in the sense that in the groups to be examined, some aspects were different, such as the diet, the absence of a group with an alternative treatment, and the introduction of a group of nondiabetic patients instead of another kind of treatment.

Another difficulty we found when comparing results is the variability when providing different results in each study, as well as the *p* values of significance, where some of them did not specify which group was the one showing the significant results.

## 5. Conclusions

HIIT could have effects on inflammatory biomarkers, based on the investigation conducted of an anti-inflammatory nature, provided that its characteristics are able to trigger the necessary impact to do so. However, due to the scarcity of available studies, it is impossible to determine which of its varieties is the best option

For having an impact on inflammatory biomarkers, HIIT should maybe trigger significant changes in the values of body composition, particularly by achieving a loss of body weight. However, for the same reason as that already mentioned, more studies are needed to allow the verification of this hypothesis.

Mainly TNFα and IL-6 are the biomarkers examined in the trials and the beneficial effect of HIIT over MCT is not clear due in part to the differences in the HIIT protocol employed. Therefore, more information is needed for other biomarkers to understand the relationship between different characteristics of HIIT. It is possible that HIIT does not trigger significant changes in every type of inflammatory biomarker, but only some of them may benefit from this type of training as a therapeutic exercise.

According to the available studies, a controlled diet may improve the effects that HIIT as a training modality might have on inflammatory biomarkers and the values of body composition.

Nowadays, it is impossible to determine whether HIIT is a better, worse or equivalent alternative to a standard aerobic training to improve the inflammatory profile. To do so, we needed more studies published to provide some answers to this mystery.

## Figures and Tables

**Figure 1 ijerph-18-12644-f001:**
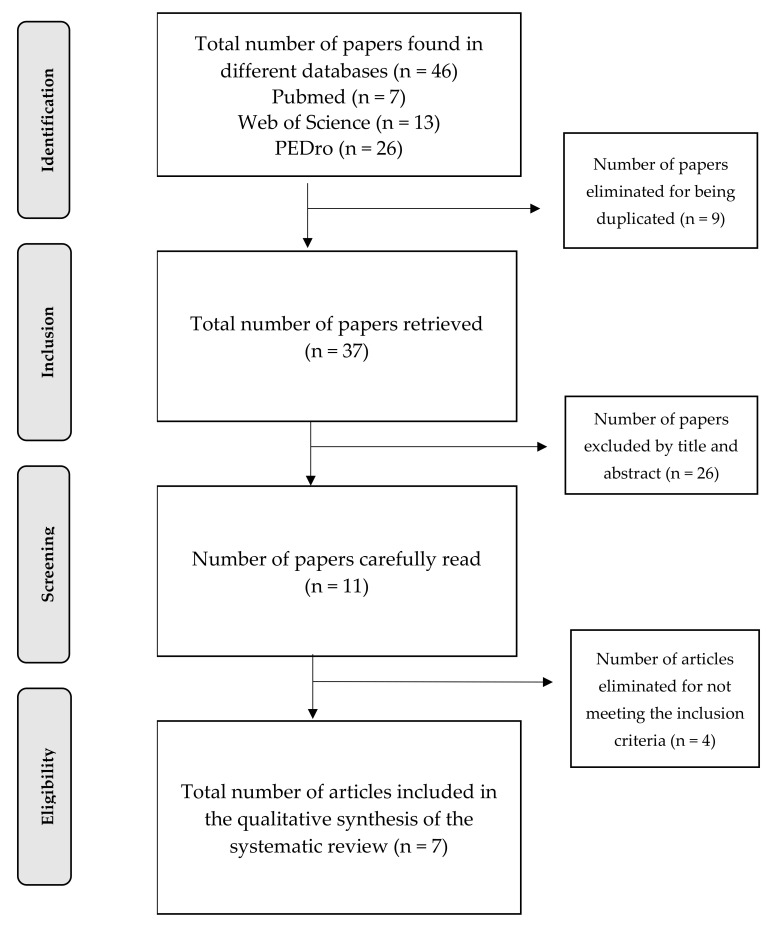
Flowchart.

**Table 1 ijerph-18-12644-t001:** Papers assessment using Downs and Black Checklist.

Downs and Black Checklist (Modified)																											
Papers	Items																										TOTAL
	1	2.	3.	4.	5.	6.	7.	8.	9.	10.	11.	12.	13.	14.	15.	16.	17.	18.	19.	20.	21.	22.	23.	24.	25.	26.	26 ITEMS
Magalhães et al., 2020	1	1	1	1	1	1	1	1	1	1	1	1	-	0	1	-	1	1	1	1	1	-	1	0	1	1	21/26
Mallard AR et al., 2017	1	1	1	1	1	1	1	1	1	0	1	-	-	0	1	-	1	1	1	1	1	-	1	0	1	-	18/26
Banitalebi E et al., 2019	1	1	1	1	1	1	1	1	0	1	1	-	-	0	1	-	1	1	1	1	1	-	1	0	0	1	18/26
Asle M et al., 2018	1	1	1	1	1	1	1	1	1	1	0	-	-	0	-	-	1	1	1	1	1	-	1	0	1	1	18/26
Dünnwald T et al., 2019	1	1	1	1	1	1	1	0	1	1	0	-	-	0	-	-	1	1	1	1	1	-	0	0	1	1	16/26
Madsen SM et al., 2015	1	1	1	1	0	1	1	1	0	1	1	1	0	0	-	-	1	1	1	1	1	-	0	0	1	-	16/26
Afrasyabi, S et al., 2019	1	1	1	1	0	1	0	1	0	0	1	-	-	0	-	-	1	0	-	1	1	-	1	0	0	-	11/26

This section may be divided by subheadings. It should provide a concise and precise description of the experimental results, their interpretation, as well as the experimental conclusions that can be drawn.

**Table 2 ijerph-18-12644-t002:** General characteristics.

Study Reference	Country	Downs and Black	Type of Study	Profile	Intervention Duration	Women/Men	Age range	Weight Conditions	Level of Activity at Study Inception	Groups Included in the Study	Intervention in Each Group
Magalhães JP et al., 2020 [50]	Portugal	21/26	RCT *HIIT vs MCT vs control (all T2D)*	T2D	12 months	Women:38/80 (47.5%) when starting. Hombres: 42/80 (52.5%) when starting.	30–75	BMI < 48 kg/m^2^	-	MCT Group N = 16	MCT Group + RT in final stages = cycle ergometer
										Control Group N = 22	Control Group = Initial recommendation on standard and non-structured physical activity in one session
										HIIT Group N = 13	HIIT Group + RT in final stages = cycle ergometer
Mallard AR et al., 2017 [52]	Australia	18/26	RCT *parallel design. HIIT vs MICT (both T2D)*	T2D	12 weeks	Women: 14/36 (38.88%) Men: 22/36 (61.11%)	44–65	HIIT = BMI 30.2 ± 2.7 kg/m2 MICT = BMI 29.6 ± 3.6 kg/m^2^	Less than 210 min/week	HIIT Group N = 20	HIIT Group: treadmill
										MICT Group N = 16	MICT Group: home based exercise
Banitalebi E et al., 2019 [28]	Iran	18/26	RCT *SIT vs Aerobic + resistance vs Control (all T2D)*	T2D	10 weeks	Women: 42 (100%)	30–65	BMI > 25 kg/m^2^	Sedentary (no more than 20 min of structured exercise of any kind for the 6 months before the study or no sprint interval exercise).	Group A + R Training N = 14	Group A + R Training: Participants free to choose cycle ergometer or treadmill.
										Control Group N = 14	Control Group: asked to keep their physical activity levels during the study.
										SIT Group N = 14	SIT Group: cycle ergometer
Asle M et al., 2018 [55]	Iran	18/26	RCT *Types of diets + HIIT vs Normal diet + HIIT (All T2D)*	T2D	24 weeks	-	36–58	BMI between 30 and 39 kg/m^2^	Sedentary (no regular exercise more than one day a week)	Experimental HIIT Group + Low carbohydrates N = 10	Experimental Group + low carbohydrates: cycle ergometer
										Experimental HIIT Group + Low fat N =10	Experimental HIIT Group + low fat: cycle ergometer
										Experimental Group = HIIT + high fat N = 10	Experimental Group = HIIT + high fat: cycle ergometer
										Control HIIT Group + Normal diet N = 9	Control HIIT Group + normal diet: cycle ergometer
Dünnwald T et al., 2019 [53]	Austria	16/26	CT *HIIT vs CMT (all T2D)*	T2D	4 weeks	-	50–65	HIIT = BMI 27.8 ± 2.8 kg/m^2^ CMT = BMI 31.8 ± 4.6 kg/m^2^	-	HIIT Group N = 8	HIIT Group: cycle ergometer
										CMT Group N = 6	CMT Group: cycle ergometer
Madsen SM et al., 2015 [54]	Denmark	16/26	CT *T2D vs Healthy (all doing HIIT)*	T2D and Healthy	8 weeks	Women 15/23 (65.22%) Hombres 8/23 (34.78%)	52 ± 2	-	-	Experimental T2D Group N = 10	Experimental T2D Group N = 10
										Healthy Control Group N = 13	Healthy Control Group N = 13
Afrasyabi S et al., 2019 [51]	Iran	11/26	RCT *HIIT vs Control* *(In both groups there are T2D and healthy subgroups. 8 groups in total)*	T2D and Healthy	12 weeks	-	40 ± 10	BMI ≥ 30 kg/m2 for obese ≤ 20 kg/m^2^ for slim	Less than moderate intensity exercise > 1.5 h per week	HIIT(O-ND-T) N = 10 HIIT (O-D-T) N = 10 HIIT (N-ND-T) N = 10 HIIT (N-D-T) N = 10 Control (O-ND-C) N = 10 Control (O-D-C) N = 10 Control (N-ND-C) N = 10 Control (N-D-C) N = 10	HIIT (O-ND-T) = Running HIIT (O-D-T) = Running HIIT (N-ND-T) = Running HIIT (N-D-T) = Running Control (O-ND-C) = Running Control (O-D-C) = Running Control (N-ND-C) = Running Control (N-D-C) = Running

RCT Randomised Clinical Trial; HIIT High intensity Interval Training; T2D Type 2 Diabetes Mellitus; BMI Body Mass Index; min minutes; kg/m2 Kilograms per square meter; h hours; SIT Short sprint Interval training; A + R Training Combined aerobic and resistance training; MCT Moderate continuous training; MICT Moderate intensity continuous training; CMT moderate continuous training; RT resistance training; O-ND-T Obesity Non Diabetic Training; O-D-T Obesity diabetic Training; N-ND-T Non Diabetic Training; N-D-T Normal Weight Diabetic Training; O-ND-C Obesity Non Diabetic Control; O-D-C Obesity diabetic control; N-ND-C Normal weight non diabetic control; N-D-C normal weight diabetic control.

**Table 3 ijerph-18-12644-t003:** HIIT characteristics and anthropometric values and biomarkers studied.

**Study Reference**	**HIIT Group Size**	HIIT Intensity	HIIT Duration (Weeks)	Frecuency (Session/Week)	Series	Working-Out Time (Sec)	Rest Time (Sec)	Working Intervals	Volume per Session	Training Periods	Body Composition	Inflammatory Biomarkers
Magalhães JP et al., 2020 [50]	13/51	70% to 90%Hrmax	52 weeks	3	1	120 s to 60 s	60 s	LI-HIIT) to (MI-HIIT	LV-HIIT	LT-HIIT	BMI (kg/m2) WC Waist circumference (cm) WBFI whole-body fat index (kg/m2) AFI android fat index (kg/m2) Weight (kg)	TNF-α IL-6, CRP sCD163
Mallard AR et al., 2017 [52]	16/36	90%–95% Hrmax	12 weeks	3	4	240 s	180 s	LI-HIIT	HV-HIIT	LT-HIIT	BMI (kg/m2) Waist circumference (cm)	TNF-α IL-8 IL-6 IL-10
Banitalebi E et al., 2019 [28]	14/42	All out	10 weeks	3	4	30 s	120 s at 50W	SI-HIIT	MV-HIIT	MT-HIIT	BMI (kg/m2) WC Waist circumference (cm) Weight (kg) LBM lean body mass (kg) Body fat (%)	Irisin IL-15 FGF21 IL-6 ANGPTL4 SPARC
Asle M et al., 2018 [55]	39/39	75% to 90% Hrmax	24 weeks	3	10	60 s	60 s 30% Hrmax	SIT-HIIT	LV-HIIT	LT-HIIT	BMI (kg/m2) WC Waist circumference (cm) Weight (kg) Height (cm)	TNF-α IL-6, Leptin Resistin Adiponectin FGF21
Dünnwald T et al., 2019 [53]	8/14	90%–95% Hrmax	4 weeks	3	5	240 s	180 s (70%Hrmax)	LI-HIIT	HV-HIIT	ST-HIIT	IMC (kg/m2) Adipose Tissue Mass (kg) Fat-free mass (kg)	TNF-α Leptin Adiponectin Irisin
Madsen SM et al., 2015 [54]	10/23	90% Hrmax	8 weeks	3	10	60 s	60 s active recovery	MI_HIIT	HV-HIIT	MT-HIIT	Weight (kg) Abdominal fat (kg)	TNF-α IL-6, IL-1, Leptin CRP
Afrasyabi S et al., 2019 [51]	40/80	85%–95% Hrmax	3 week 12 weeks	3	6 to 12	60 s	60 s	MI_HIIT	HV-HIIT	LT-HIIT	BMI (kg/m2) Weight (kg) Height (cm)	TNF-α

Hrmax Maximum heart rate; HIIT High Intensity Interval Training; All out; s seconds; W watts; SIT short-term; SI-HIIT short-interval; MI-HIIT moderate-interval; LI-HIIT long-interval; LV-HIIT low-volume; MV-HIIT moderate-volume; HV-HIIT high-volume; ST-HIIT short-term; MT moderate-term; LT long-term; BMI Body Mass Index; WC waist circumference; WBFI whole-body fat index; AFI android fat index; LBM lean body Mass; TNF-α Tumoral necrosis factor α; IL-6 Interleukin 6; IL-8 Interleukin 8; IL-10 Interleukin 10; IL-15 Interleukin 15; IL-1 Interleukin CRP C-reactive protein; sCD163 macrophage activation marker-soluble CD163; ANGPTL4 Angiopoietin 4; FGF 21; SPARC Secreted protein acidic and rich in cysteine.

**Table 4 ijerph-18-12644-t004:** HIIT characteristics and anthropometric values and biomarkers studied.

Paper	Results
Magalhães et al., 2020 [50]	No significant results found for the HIIT group. However, there was a significant reduction in the MCT (−3.6 ± 16.4 pg/mL) Group compared with the control (7.0 ± 17.3 pg/mL) (p = 0.047) for the IL-6. There is no information on post-treatment to determine if they recorded significant changes for the body composition variables.
Mallard AR et al., 2017 [52]	No significant results were found (p < 0.05) for biomarkers IL-10, IL-6, IL-8, TNF-α within any of the intervention groups (HIIT and MCT) throughout 12 weeks. There is no information on post-treatment to determine if they recorded significant changes for the body composition variables.
Banitalebi E et al., 2019 [28]	A significant reduction was recorded for biomarkers IL-15 (p = 0.02) and IL-6 (p = 0.002) after the treatment period in each of the intervention groups; it was greater within the SIT group (−0.23 and −0.67 mean difference) than in those of A+R (−0.21 and −0.52 mean difference) and control (−0.07 and −0.23 mean difference). Significant differences recorded among different groups after having received treatment for biomarker Irisin (SIT = 42.15, A + R = 68.57, Control = 17.15 ng/mL mean difference) (p = 0.009). Regarding the values of body composition, significant differences were found among groups for BMI (p = 0.01), weight (p = 0.02), LBM (p = 0.01) values.
Asle M et al., 2018 [55]	The TNF-α showed a significant reduction after intervention (p < 0.001) and among groups (p < 0.001). (Low CHO-18.69, Low Fat-48.06, High Fat 0.66, and Control 11.69 percentage of change)The IL-6 did not show a significant reduction after intervention, but it did show it among groups (p < 0.001). (Low CHO-32.10, Low Fat-24.97, High Fat-18.67, and Control-4.23 percentage of change)The leptin showed a significant reduction after intervention (p < 0.001) and among groups (p < 0.001). (Low CHO-53.92, Low Fat-30.26, High Fat 0.11, and Control-1.99 percentage of change)Adiponectin showed a significant increase after intervention (p < 0.001) and among groups (p< 0.001). (Low CHO 18.10, Low Fat-42.32, High Fat 1.84, and Control 4.89 percentage of change) The FGF21 showed a significant increase after intervention (p < 0.006) and among groups (p < 0.001). (Low CHO 55.86, Low Fat 52.30, High Fat-13.43, and Control 21.66 percentage of change)Resistin: it showed a significant reduction after intervention (p < 0.006) and among groups (p < 0.001). (Low CHO-28.29, Low Fat-15.27, High Fat-9.18, and Control-1.30 percentage of change)Reductions were more remarkable in groups with low carbohydrates and low-fat diet.There is no information on post-treatment to determine if they recorded significant changes for the body composition variables.
Dünnwald T et al., 2019 [53]	A significant increase was recorded within the HIIT group for the biomarker irisin (0.57 ± 0.04 pre, 0.61 ± 0.06 post (ug/mL) compared to CMT group (0.60 ± 0.06 pre, 0.58 ± 0.07 post) (p ≤ 0.05). Moreover, a significant increase was recorded within the CMT group for the fat-free mass anthropometric variable (p ≤ 0.05).
Madsen SM et al., 2015 [54]	HIIT did not lead to significant variations in inflammatory biomarkers in patients with T2D. However, a significant increase was observed in the group of healthy people with respect to the IL-1Ra biomarker (p = 0.03). In relation to the values of body composition, a significant reduction in weight (p < 0.01) and abdominal fat mass (p < 0.01) was observed after treatment within the T2D group and in weight (p < 0.001) and abdominal fat mass (p < 0.05) within the control group (healthy subjects).
Afrasyabi S et al., 2019 [51]	The TNF-α was significantly diminished after treatment with HIIT (p = 0.001) in groups N-ND-T, O-D-T and O-ND-T, and there were significant differences among groups, the O-D-T recording the greatest reduction (p = 0.001).There is no information on post-treatment to determine if they recorded significant changes for the body composition variables.

HIIT High Intensity Interval Training; T2D Type 2 Diabetes Mellitus; MCT Moderate continuous training; CMT Moderate continuous training; TNF-α Tumoral necrosis factor α; IL-6 Interleukin 6; IL-15 Interleukin 15; FGF21; N-ND-T normal weight, nondiabetic training; O-D-T obesity diabetic training; O-ND-T obesity non diabetic training.

**Table 5 ijerph-18-12644-t005:** HIIT Classification by Wen D et al. [44].

Working Out Interval	Close to Maximum (All-Out) Intensity: ≥ 90 VO_2max_ / ≥ 95% Hrmax / ≥120% v/pVO_2max_		
	≤10s RST	10–30s (SIT)	
	Second Máximum Intensity: 80–90% VO_2max_ / 85–85% Hrmax / 90–120% v/pVO_2max_		
	≤ 30 s (SI-HIIT)	30 s to 2 min (MI-HIIT)	≥ 2 min (LI-HIIT)
Volume per Session (Duration x Repetition)	≤5 min (LV-HIIT)	5 to 15 min (MV-HIIT)	≥15 min (HV-HIIT)
Training Period (Intervention Duration)	≤4 weeks (ST-HIIT)	4 to 12 weeks (MT-HIIT)	≥12 weeks (LT-HIIT)

HIIT High-Intensity Interval Training; RST repeated-sprint; SIT short-term; SI-HIIT short-interval; MI-HIIT moderate-interval; LI-HIIT long-interval; LV-HIIT low-volume; MV-HIIT moderate-volume; HV-HIIT high-volume; ST-HIIT short-term; MT moderate-term; LT long-term; VO2max Maximum Oxygen Volume; Hrmax Maximum Heart Rate; v/pVO_2_max velocity/power at VO_2_max. min minutes; s seconds; weeks.

## Data Availability

Data is contained within the article or Appendix A.

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
