# Peer review of "Effects of High-Intensity Interval Training on Inflammatory Biomarkers in Patients with Type 2 Diabetes. A Systematic Review"

_ijerph, 2021, doi:10.3390/ijerph182312644_

Round 1
Reviewer 1 Report
This systematic review deals with an important and relevant topic for all community and reveals a great importance in terms of practical application. I enjoyed reading this review, and so I congratulate the authors. Minor repairs are carried out in order to increase the quality of the review.
Page 2 Line 75 - The HIIT is very popular. However, the saying that this is the most effective training for cardiorespiratory and metabolic levels is exaggerated. The difference between continuous and interval aerobic training and HIIT on cardiovascular capacity is the exercise's time. So HIIT is time/efficient better than continuous and interval aerobic exercise.
Page 3, Line 139 - I miss in this paragraph an attempt to explain how HIIT positively interfere with the inflammatory process.
Page 4 Line 174 - Why did you not use the Scopus database?
You did not think it pertinent to ask, at the end of the selection of articles, the openião of experts in the area.
Figure 1 - Please discriminate the papers found in each database.
Table 4: - It is not express the HIIT intensity in Madsen SM et al. (2015). Please provide if possible; - In the column "HIIT duration (weeks) and frequency (days), please change the order of the values. Firstly you have the week frequency and after the duration in weeks.
-
Author Response
Reviewer 1:
General comment: This systematic review deals with an important and relevant topic for all community and reveals a great importance in terms of practical application. I enjoyed reading this review, and so I congratulate the authors. Minor repairs are carried out in order to increase the quality of the review.
Minor comments:
Author comment (AC): Page 2 Line 75 - The HIIT is very popular. However, the saying that this is the most effective training for cardiorespiratory and metabolic levels is exaggerated. The difference between continuous and interval aerobic training and HIIT on cardiovascular capacity is the exercise's time. So HIIT is time/efficient better than continuous and interval aerobic exercise.
Author Answer (AA): Thanks for the comment. We have tried to be more cautious with the comment, stating that it is one of the most popular and effective training modalities for improving both at cardiac respiratory and metabolic levels.
We have also added the following sentence:
“Compared to traditional continuous aerobic training, which is characterized by exercise sessions of long duration at moderate intensities, HIIT appears also to be an effective tool to improve health but in shorter sessions [1,2].”
AC: Page 3, Line 139 - I miss in this paragraph an attempt to explain how HIIT positively interfere with the inflammatory process.
AA: Thanks for the comment. We have included the next text in agreement with the reviewer 1 in the page 3:
“Regular exercise exerts an anti-inflammatory chronic effect diminishing the plasmatic markers of inflammation [3]. The elevation of circulating anti-infammatory biomarkers after an acute bout of exercise seems to be dependent of intensity, with greater response after HIIT compared to moderate-intensity continuous exercise [4,5]. The HIIT relies more heavily on carbohydrate oxidation than endurance training which has been related with a blunted acute inflammatory response [6,7]. However, this finding has not consistently been shown in the literature [8,9].”
AC: Page 4 Line 174 - Why did you not use the Scopus database?
AA: According to "Chapter 4: Searching for and selecting studies" of the "Cochrane Handbook for Systematic Review of Interventions" version 6.2, 2021, Scopus is not considered as one of the main databases in which to search for clinical trials of interest. We consider it to be a bibliographic database that collects citations received by articles, but not a primary source within the field of biomedicine [10].
AC: You did not think it pertinent to ask, at the end of the selection of articles, the opinion of experts in the area.
AA: Thank you very much and your contribution seems very important to us, however, we have tried to be as rigorous as possible in the review systematization process, in accordance with the perspective of the PRISMA checklist [11], where expert opinions cannot be included. Being the main reason why they have not been included.
AC: Figure 1 - Please discriminate the papers found in each database.
AA: The following items have been included in the Figure 1: Pubmed (n=7), Web of Science (n=13), PEDro (n=26).
AC: Table 4: - It is not express the HIIT intensity in Madsen SM et al. (2015). Please provide if possible; - In the column "HIIT duration (weeks) and frequency (days), please change the order of the values. Firstly, you have the week frequency and after the duration in weeks.
AA: Thanks for the comment. We have added this information in the table. In addition, we have modified and included editable tables.

Reviewer 2 Report
Thank you very much for your interesting manuscript. However, I have some minor comments for this manuscript.
Minor comments
- I would suggest the information regarding with the table 5 “HIIT characteristics and anthropometric values and biomarkers studied”. This table should be presented with more precise and clear information rather than the current style.
- It is really interesting to report how is the generalizability of your findings to reveal it at the end of discussion.
Author Response
Reviewer 2:
General comment: Thank you very much for your interesting manuscript. However, I have some minor comments for this manuscript.
Minor comments:
AC: I would suggest the information regarding with the table 5 “HIIT characteristics and anthropometric values and biomarkers studied”. This table should be presented with more precise and clear information rather than the current style.
AA: We have completed the table with information more precise and clear. In fact, we have included quantitative data on the effect of HIIT on the variables analysed in the different studies.
AC: It is really interesting to report how is the generalizability of your findings to reveal it at the end of discussion.
AA: To provide this information, we also have included the total number of participants included in all the studies in this systematic review, as well as the biomarkers analysed in each study.

Reviewer 3 Report
This paper is clearly written and well organised. The introduction and background are reasonable given the premise of the paper. Figures and tables are comprehensive and helpful .
As a reviewer, I would like the author to provide/clarify more information on:
1) The effect of healthy diet in patients with type 2 diabetes
2) The relationship between healthy diet vs HIIT
3) Please define 'a healthy diet' and 'low carb'. I understand that low carb diet plays the major roles in insulin level as well (section 4.4) .
4) Please define 'HIIT'. I understand that each particular study had different HIIT setting ( Table 3).
5) What is the best biomarker ( Table 4)
Author Response
Reviewer 3:
General comment: This paper is clearly written and well organised. The introduction and background are reasonable given the premise of the paper. Figures and tables are comprehensive and helpful.
Minor comments:
As a reviewer, I would like the author to provide/clarify more information on:
AC: The effect of healthy diet in patients with type 2 diabetes.
AA: We have included the following sentence:
“Although more long-term intervention trials are needed, Papamichou et al. [12] have showed the effect that dietary patterns of a healthy diet should be implemented in public health strategies in order to better control glycaemic markers in patients with T2D.”
AC: The relationship between healthy diet vs HIIT.
AA: We have included the following sentence:
“The low-carbohydrate high-fat diet is a re-emerged dietary approach, which is characterized by decreased carbohydrate intake and high levels of fat consumption with adequate protein provided [13]. A recent systematic review and meta-analysis has showed low-carbohydrate high-fat diet combined with HIIT reduces body weight and fat mass while maintaining lean body mass and enhancing aerobic capacity [14].”
AC: Please define “a healthy diet” and “low carb”. I understand that low carb diet plays the major roles in insulin level as well (section 4.4).
AA: We have included the following information:
“A healthy diet is a diet according with recommendations from authoritative organizations [15]. “
Low carbohydrate diet have been defined in the previous comment. Evidence of low-carb diet and T2D included, it has been included in the previous comment too.
AC: Please define 'HIIT'. I understand that each particular study had different HIIT setting (Table 3).
AA: We have modified Tables 3 and 4, so that the characteristics of HIIT are clearer. Having remained these information in table 4.
AC: What is the best biomarker (Table 4)?
AA: We have include this information in conclusions section.
“Mainly TNFα and IL-6 are the biomarkers examined in the trials and is no clear the beneficial effect of HIIT over MCT due in part to the differences in HIIT protocol employed. Therefore, more information is needed for other biomarkers to understand the relationship with different characteristic of HIIT.”
